# Environmental Stimulation Counteracts the Suppressive Effects of Maternal High-Fructose Diet on Cell Proliferation and Neuronal Differentiation in the Dentate Gyrus of Adult Female Offspring via Histone Deacetylase 4

**DOI:** 10.3390/ijerph17113919

**Published:** 2020-06-01

**Authors:** Wen-Chung Liu, Chih-Wei Wu, Pi-Lien Hung, Julie Y. H. Chan, You-Lin Tain, Mu-Hui Fu, Lee-Wei Chen, Chih-Kuang Liang, Chun-Ying Hung, Hong-Ren Yu, I-Chun Chen, Kay L.H. Wu

**Affiliations:** 1Division of Plastic and Reconstructive Surgery, Kaohsiung Veterans General Hospital, Kaohsiung 81362, Taiwan; dliu@vghks.gov.tw (W.-C.L.); lwchen@vghks.gov.tw (L.-W.C.); 2Department of Surgery, School of Medicine, National Yang-Ming University, Taipei 11221, Taiwan; 3Institute for Translational Research in Biomedicine, Kaohsiung Chang Gung Memorial Hospital, Kaohsiung 83301, Taiwan; 2106131184@nkust.edu.tw (C.-W.W.); jchan@cgmh.org.tw (J.Y.H.C.); tainyl@adm.cgmh.org.tw (Y.-L.T.); jectry@cgmh.org.tw (C.-Y.H.); harry741225@cgmh.org.tw (I.-C.C.); 4National Kaohsiung University of Science and Technology, Kaohsiung 83301, Taiwan; 5Department of Pediatrics, Kaohsiung Chang Gung Memorial Hospital and Chang Gung University College of Medicine, Kaohsiung 83301, Taiwan; flora1402@cgmh.org.tw (P.-L.H.); yuu2002@cgmh.org.tw (H.-R.Y.); 6College of Medicine, Chang Gung University, Kaohsiung 83301, Taiwan; 7Department of Neurology, Kaohsiung Chang Gung Memorial Hospital and Chang Gung University College of Medicine, Kaohsiung 83301, Taiwan; k8601085@cgmh.org.tw; 8Center for Geriatrics and Gerontology and Division of Neurology, Kaohsiung Veterans General Hospital, Kaohsiung 81362, Taiwan; ckliang@vghks.gov.tw; 9Department of Senior Citizen Services, National Tainan Institute of Nursing, Tainan 700, Taiwan

**Keywords:** maternal high-fructose diet, neural stem cell proliferation, neuronal differentiation, histone deacetylase 4, enriched housing

## Abstract

Maternal high-fructose diets (HFD) impair the learning and memory capacity of adult female offspring via histone deacetylase 4 (HDAC4). Hippocampal adult neurogenesis is important for supporting the function of existing neural circuits. In this study, we investigated the effects of maternal HFD on hippocampal neural stem cell (NSC) proliferation and neuronal differentiation in adult offspring. Increased nuclear HDAC4 enzyme activity was detected in the hippocampus of HFD female offspring. The Western blot analyses indicated that the expressions of sex-determining region Y box2 (SOX2) and the transcription factor Paired Box 6 (PAX6), which are critical for the progression of NSC proliferation and differentiation, were downregulated. Concurrently, the expression of Ki67 (a cellular marker for proliferation) and doublecortin (DCX), which are related to NSC division and neuronal differentiation, was suppressed. Intracerebroventricular infusion with class II HDAC inhibitor (Mc1568, 4 weeks) led to the upregulation of these proteins. Environmental stimulation reversed the expression of Ki67 and DCX and the counts of Ki67- and DCX-positive cells in the hippocampi of HFD offspring as a result of providing the enriched housing for 4 weeks. Together, these results demonstrate that the suppressive effects of maternal HFD on hippocampal NSC proliferation and neuronal differentiation are reversibly mediated through HDAC4 and can be effectively reversed by environmental stimulation. The advantageous effects of environmental enrichment were possibly mediated by HDAC4 suppression.

## 1. Introduction

The health status of individual adults can be permanently shaped by the nutritional support received during fetal and early postnatal life [1,2]. Fructose is a common sweetener in the daily diet, found in such foods as desserts and beverages. Overconsumption of fructose has been a public health issue since the 1970s [3,4]. Notably, maternal high-fructose diets (HFDs) during gestation and lactation shape the brain functions of adult offspring; this has been demonstrated in both animal and human studies [5,6]. For example, maternal HFD disrupts brain development, resulting in a decrease in the brain size of adult offspring [7]. Moreover, adult female offspring who were associated with maternal HFD demonstrated lower performance in terms of learning and memory [6]. However, the underlying mechanisms remain largely unknown. Adult neurogenesis in the dentate gyrus of the hippocampus plays important roles in supporting and maintaining the functions of learning and memory by generating new neurons to replace impaired neurons in the existing neural circuits [8,9,10,11]. Whether maternal HFD impairs adult neurogenesis in female offspring needs thorough investigation.

In the adult hippocampus, neural stem cells (NSCs) are the source of progenitor cells residing in the subgranular zone (SGZ) of the dentate gyrus (DG). They are involved in the process of generating new neurons, which is known as adult neurogenesis [8,12]. Several processes take place during the progression of adult neurogenesis, including proliferation, differentiation, maturation, and survival of the newborn neurons [13]. Several stage-specific markers have been identified. For example, Nestin is an intermediate filament of radial glia-like NSCs. Therefore, it has been frequently used as a marker of NSCs [14]. The transcription factor sex-determining region Y box 2 (SOX2) is highly expressed in type 1a and 1b neural progenitor cells (NPCs) and is critical for self-renewal, proliferation, and differentiation [15,16,17]. The transcription factor Paired Box 6 (PAX6) is a key regulator in the promotion of the transition of NSCs to NPCs via interaction with SOX2 [18]. Ki67, a common marker for proliferation, is a nuclear protein that is highly expressed in dividing cells, while doublecortin (DCX) is expressed in immature newborn neurons [19]. These newborn neurons have been functionally linked to learning and memory [9], and act by replacing impaired neurons to maintain the functional neural circuits [11,20]. The enhancement of adult neurogenesis improves the performance of learning and memory [21,22]. On the contrary, a decrease in adult neurogenesis impairs memory [10,23] and cognitive flexibility [24,25,26]. The processes of adult neurogenesis are highly sensitive to the nutrient supply. For instance, the overconsumption of fructose results in the suppression of NSC proliferation, neuron differentiation, and maturation in the dentate gyrus [27]. Whether the negative effect of fructose extends to the offspring is largely unknown.

Histone acetylation contributes to the modulation of neurogenesis [28]. At least five classes of histone deacetylases (HDACs; HDAC class I, IIa, IIb, III and IV) have been determined to be involved in the regulation of gene expression in learning and memory by modulating the status of histone acetylation [28,29]. HDAC4 is a member of Class IIa HDACs, which can shuttle from the cytoplasm to the nucleus for the repression of gene expression [29,30]. A previous study indicated that maternal HFD induces HDAC4 to suppress the expression of brain-derived neurotrophic factor (BDNF) [6], a growth factor critical for the maintenance of adult hippocampal neurogenesis [31]. Despite the impressive progress in the study of HDACs (Reviewed by Yao et al. [32]), less attention has been paid to the role of HDAC4 in adult neurogenesis.

Environmental enrichment has been demonstrated to increase adult neurogenesis [33] and, thus, to increase the number of hippocampal neurons [34]. The underlying mechanism has been linked to the increment of hippocampal BDNF [35]. Functionally, the stimulation of environmental enrichment enhances the capacity of learning and memory [36,37,38]. In addition, an increasing amount of evidence indicates that environmental enrichment might increase histone acetylation [39] via the suppression of the HDAC4 functions [6,40]. However, whether the environmental stimulation is able to rewrite the HDAC4-associated suppression of adult neurogenesis in the later life of HFD offspring is still unknown.

Our previous study demonstrated that maternal HFD suppresses hippocampal BDNF expression and decreases the ability of learning and memory via HDAC4 in adult female offspring. In our current study, the levels of hippocampal NSC proliferation and neuronal differentiation were characterized in three-month-old female offspring to elucidate the effect of maternal HFD on hippocampal adult neurogenesis. To investigate the contribution of HDAC4 to maternal HFD-altered adult neurogenesis, the levels of various markers of NSC expression, namely Nestin (a marker for radial glia-like NSCs), SOX2 (a marker for NSC self-renewal), Ki67 (a marker for division), PAX6 (a marker for the differentiation from NSCs to NPCs) and DCX (a marker for neuronal differentiation), were assayed in response to treatments with the class II HDAC inhibitor, Mc1568. The use of enriched housing, through the addition of toys and nesting material, was also evaluated to assess its capacity to reverse this observed maternal HFD-induced impairment.

## 2. Materials and Methods

### 2.1. Animals

Seven-week old nulliparous female (n = 8) and male (n = 8) Sprague–Dawley (SD) rats were purchased from the BioLASCO Taiwan Co., Ltd., Taipei, Taiwan. All rats were acclimatized in a room with controlled temperature (22 ± 1 °C), humidity (55 ± 5%), and light (12:12 light–dark cycle, with light on from 05:00) conditions within the Association for Assessment and Accreditation of Laboratory Animal Care (AAALAC)-certified animal facility for at least 14 days prior to the commencement of experiments. All experiments were carried out in accordance with the guidelines for animal experimentation endorsed by the institutional animal care and use committee (IACUC) of the Kaohsiung Chang Gung Memorial Hospital. Each male rat was housed with a single female, and the occurrence of mating was confirmed by the presence of a vaginal plug. Pregnant female rats were randomly assigned to receive a regular chow (ND; 5001, 3.35 kcal/g; Purina, USA) or high-fructose diet (HFD; TD. 89247, 60% fructose; 3.6 kcal/g; Envigo, UK) during the periods of pregnancy and lactation (the details of dietary contents are listed in Table 1). After weaning (3 weeks old), all offspring were fed with ND. At 8 weeks old, female offspring from the ND and HFD groups were randomly assigned to (1) be treated with Mc1568 (5 μM) intracerebroventricular infusion (icv) for 4 weeks or (2) be housed in a regular cage or enriched housing (the details of the scheme are plotted in Figure 1). Both food and water were provided ad libitum. All groups of offspring received regular chow after weaning.

### 2.2. Enriched Housing

At the age of 8 weeks, female offspring from both the ND and HFD groups were randomly assigned to the following groups: ND in a standard cage (ND, n = 12); HFD in a standard cage (n = 12); and HFD in an environmentally enriched cage (HFD + enriched housing (En), n = 6) for 4 weeks. Three rats were kept in each cage. The environmentally enriched cage was a standard cage (10.5 × 19 × 18 inches) equipped with plastic toys and nesting material. To maintain novelty, the rats were transferred to an alternative new cage with new toys and nesting material every week. Food and water were provided ad libitum.

### 2.3. Intracerebroventricular (icv) Infusion

Female offspring which been randomly assigned for Mc1568 icv infusion were placed in a sealed Plexiglas box in which 4% isoflurane, and anesthesia were introduced using a 2 L/min oxygen flow. Rats were then placed in a standard stereotaxic device equipped with a gas anesthesia nose cover to maintain anesthesia throughout surgery with 2% isoflurane and a 600 mL/min oxygen flow. The Mc1568-filled osmotic pump was surgically implanted with an ALZET Brain Infusion probe (Alzet 1002) into the right lateral ventricle, with the tip of the infusion probe at the following coordinates with reference to the Bregma: anterior/posterior −1.4 mm; medial/lateral 1.8 mm; dorsal/ventral −3.0 mm (Paxinos and Watson, 1986). An intracerebroventricular infusion of the treating chemicals was carried out for 28 days. Control infusions of artificial cerebrospinal fluid (aCSF) served as the volume and vehicle control. ND, n = 12; HFD, n = 12; HFD+Mc1568 (HFD+Mc), n = 6.

### 2.4. Total Protein Isolation

Offspring were deeply anesthetized at 3 months old by sodium pentobarbital (100 mg/kg; Sigma, St. Louis, MO, USA) and were transcardially perfused with saline. The skull was removed to expose the brain. The sagittal dissection from the middle line was performed to expose and to obtain the hippocampi. For Western blotting and ELISA analyses, the fresh hippocampus was obtained after perfusion. The sampled hippocampus was homogenized with a Dounce grinder with a tight pestle in ice-cold lysis buffer (Sigma-Aldrich) containing a protease inhibitor cocktail (Sigma-Aldrich) to prevent protein degradation. The concentration of the total protein extracted was estimated using a Micro Bicinchoninic Acid (BCA) Protein Assay kit (Thermo Fisher Scientific Inc., Waltham, MA, USA).

### 2.5. Nuclear Protein Extraction

To evaluate the activity of nuclear HDAC4, nuclear and cytosolic proteins were separately extracted. Tissue samples isolated from the hippocampus were homogenized using a Dounce grinder with a loose pestle in ice-cold CelLytic™ MT Cell Lysis Reagent (Sigma-Aldrich) containing a protease inhibitor cocktail (Sigma-Aldrich). The homogenate was centrifuged at 2000× *g* for 10 min and the supernatant was collected as a cytosolic fraction. The pellet was washed with ice-cold PBS twice and then resuspended in lysis buffer (Sigma-Aldrich). To harvest the nuclear proteins, the pelleted nuclei were resuspended in 15–20 μL extraction buffer (Sigma-Aldrich) and incubated on ice for 2 h to rupture the nuclear membrane. The nuclear suspension was centrifuged at 14,000× *g* for 30 min at 4 °C, and the supernatant was saved as the nuclear protein for further analyses. The purity of protein from the nuclear and cytosolic fractions was verified by assessing the expression of markers, TATA-binding protein (TBP, a transcription factor that binds specifically to a DNA sequence named the TATA box; 1:1000, 8515, Cell Signaling Technology Inc., Danvers, MA, USA) and β-actin (1:10,000, AB8226, Abcam, Cambridge, UK), respectively. Protein concentration was determined using a Micro BCA Protein Assay kit (Thermo Fisher Scientific Inc.).

### 2.6. Histone Deacetylase 4 Activity Assay

The extracted nuclear proteins (200 μg/sample) were incubated with HDAC4 primary antibody (1:100, sc-11418x, Santa Cruz Biotechnology Inc., Dallas, TX, USA) to extract and enrich the HDAC4 for enzyme activity assay. After immunoprecipitation by incubating at 4 °C overnight, the isolated nuclear HDAC4 was prepared for the HDAC enzyme activity assay in a 96-well plate by following the guidelines (K331, BioVision Inc.). In short, the prepared samples, as well as the positive and negative controls were loaded into the individual wells at 85 µL/well. Then, 10 μL of 10x HDAC Assay Buffer was then applied to each well followed by the addition of the HDAC colorimetric substrate. The reaction was incubated at 37 °C for 1 h. Lysine Developer was then added with incubation at 37 °C for 0.5 h to stop the reaction. The colorimetric signals were read in an ELISA plate reader (Thermo Fisher Scientific Inc.) at 400 or 405 nm. HDAC activity can be expressed as the relative O.D. value per µg protein sample. The positive control provided by the kit was the nuclear extract of the HeLa cells, while the prepared samples with added trichostatin were adopted as negative controls. The protein concentration was determined by a Micro BCA Protein Assay kit (Thermo Fisher Scientific Inc.).

### 2.7. Western Blot Analysis

Protein expression in the hippocampus was separated using 10–12% sodium dodecyl sulfate polyacrylamide gel electrophoresis (SDS-PAGE). Samples from each group contained an equivalent amount of nuclear or total protein per well. The electrophoretic proteins were transferred onto a polyvinylidene difluoride membrane (Immobilon-P membrane; Millipore; Bedford, MA, USA) and probed with specific antibodies against Ki67 (1:1000, Ab16667, Abcam), SOX2 (1:1000, Ab97959, Abcam), Nestin (1:1000, Ab6142, Abcam), PAX6 (1:1000, MAB5552, Merck Millipore, Middlesex, MA, USA), Doublecortin (DCX, 1:1000, Ab18723, Abcam), and HDAC4 (1:1000, sc-11418, Santa Cruz Biotechnology Inc.). Membranes were then incubated with the appropriate horseradish peroxidase–conjugated secondary antibody (Jackson ImmunoResearch Laboratories Inc., West Grove, PA, USA). The specific antibody–antigen complex was detected using an enhanced chemiluminescence Western Blot detection system (Thermo Fisher Bioscience). The amounts of detected protein were quantified using ImageJ software (NIH, Bethesda, MD, USA). The purity of the nuclear and total fractions was verified by assessing the expression of TBP and β-actin (Millipore), respectively.

### 2.8. Brain Tissue Processing and Immunohistochemistry Labeling

For morphological analysis, forebrains were obtained and post-fixed in 4% paraformaldehyde for 72 h at 4 °C after perfusion. Thereafter, samples were cryoprotected with 30% sucrose solution at 4 °C. The 30% sucrose solution was renewed twice in 7 days. These brain samples were then sliced with a freezing microtome at 30 μm. Brain slices were collected in cryoprotectant as previously described [6] and stored at −20 °C for further immunohistochemical staining. In short, the paraformaldehyde fixed brain sections were washed with PBS, followed by permeabilization for 5 min with 0.1% Triton X 100 in 0.1% sodium citrate. After washing, the sections were stained at room temperature overnight with mouse anti-Ki67 antibody (1:1000, Ab16667, Abcam) for newly proliferated cell or goat anti-doublecortin (DCX, 1:1000, Ab18723, Abcam) for the immature neurons. Then, the specimens were rinsed three times in PBS to wash out non-binding antibodies. Thereafter, the brain specimens were incubated with the appropriate peroxidase-conjugated secondary antibody (Vector, Burlingame, CA, USA) for 2 h at room temperature and then rinsed three times in PBS. The specimens were then incubated with an avidin–biotin peroxidase (Vector) using 3,3′diaminobenzidine as the substrate. After washing three times in PBS, the specimens were imaged using an Olympus light microscope (BX51, Tokyo, Japan).

### 2.9. Statistical Analysis

Data were expressed as means ± SEM. For biochemical experiments that involved multiple groups, one-way analysis of variance with repeated measures was used to assess the group means. This was followed by the Tukey’s multiple range test for post-hoc assessment of the individual means. The statistical software GraphPad Prism 5 was used to perform the statistical analyses. ** p* < 0.05 indicates the values that were considered statistically significant from the control (ND group) values, while *^#^ p* < 0.05 indicates the values that were considered statistically significant from the from HFD group.

## 3. Results

### 3.1. The HDAC4 Activity in the Hippocampus of Adult Female Offspring Was Increased by Maternal High-Fructose Diet

Nuclear HDAC4 negatively regulates hippocampal functions [6]. In this study (as presented in Figure 1), we evaluated the activity of nuclear HDAC4 in the hippocampus of adult female offspring from mothers with ND and HFD during gestation and lactation. The results of the enzyme activity assay indicate that the nuclear HDAC4 activity in the hippocampus was significantly enhanced in the HFD group compared with the ND group (Figure 2). Further, intraventricular infusion with the inhibitor of class II HDACs, MC-1568 (Mc) effectively suppressed the deacetylation activity of HDAC4 in the HFD+Mc group. These results suggest that the HDAC4 activity was enhanced by maternal HFD.

### 3.2. Inhibition of HDAC4 Reversed the Suppression Due to Maternal HFD as Assessed Using Markers for Hippocampal Neural Progenitor Cells

To further evaluate the role of maternal HFD-increased HDAC4 in neural stem cell (NSC) self-renewal and differentiation, the HDAC4-specific inhibitor, Mc1568 (Mc), was administered via icv for 4 weeks. The Western blot analyses indicated that the expression of SOX2, a transcription factor involved in the self-renewal and differentiation of neural progenitor cells (NPCs), was downregulated in the HFD group (Figure 3a) while the level of Nestin showed no significant changes between groups (Figure 3b). In addition, the expression of PAX6, a critical driver of the process of neurogenesis, was suppressed in the HFD group (Figure 3c). The intraventricular infusion with MC-1568 effectively enhanced the expressions of SOX2 and PAX6, which were downregulated by maternal HFD. These results suggest that the mechanism by which maternal HFD impairs NSC self-renewal and the process of differentiation in the adult hippocampus involves HDAC4 activation.

### 3.3. Inhibition of HDAC4 Reversed the Suppressed Markers of Cell Proliferation and Neuronal Differentiation in the Hippocampus of the HFD Group

Hippocampal SOX2 and PAX6, both of which are involved in adult neurogenesis, were suppressed by maternal HFD-activated HDAC4 in adult female offspring, implying a decrease in NSC proliferation and neuronal differentiation. To further evaluate the role of the increased HDAC4 activation in the NSC proliferation and differentiation, the HDAC4-specific inhibitor, Mc1568 (Mc), was administered by icv infusion for 4 weeks. The results of Western blots indicate that the suppression of Ki67 in the hippocampus of the HFD group was effectively reversed by MC-1568 treatment (Figure 4a). Moreover, the decrement of DCX expression in the HFD group was enhanced by the application of MC-1568 (Figure 4b). These results support the fact that HDAC4 activation, induced by maternal HFD, is involved in the suppression of adult neurogenesis—a process which requires functional cell proliferation and neuronal differentiation.

### 3.4. Enriched Housing Suppressed HDAC4 Activity in the Hippocampus of Adult Female Offspring Exposed to a Maternal High-Fructose Diet

Environmental enrichment has been demonstrated to inhibit HDAC4 activity [6]. The activity assays of nuclear HDAC4 indicated that the enhanced HDAC4 activity in the HFD group was effectively suppressed by enriched housing (Figure 5). These results suggested that maternal HFD-induced HDAC4 activation in the hippocampus can be suppressed by environmental enrichment.

### 3.5. Enriched Housing Reversed the Expression of Hippocampal Ki67 in Adult Female Offspring Suppressed by Maternal High Fructose

The cell count results demonstrate that, in the HFD offspring, enriched housing effectively reversed the numbers of Ki67-positive cells in the dentate gyrus of the HFD group, which was suppressed by maternal HFD (Figure 6A lower right panel). The representative immunohistochemical images of the dentate gyrus also demonstrate that fewer Ki67-positive cells were detected in the HFD group in comparison with the ND group, while there were more Ki67-positive cells in the dentate gyrus of the HFD+En group (Figure 6a). Western blot analyses further indicated that the expression of the Ki67 protein was increased in the hippocampus of the HFD+En group in comparison with the HFD group (Figure 6b).

### 3.6. Enriched Housing Reversed the Suppression of Hippocampal DCX Related to Maternal High-Fructose Diet in Adult Female Offspring

The cell count results of DCX-positive cells in the dentate gyrus demonstrated that, in the HFD offspring, enriched housing effectively reversed the numbers of DCX-positive cells in the dentate gyrus, which was suppressed by maternal HFD (Figure 7a lower right panel). The representative immunohistochemical images of the dentate gyrus also showed that more DCX-positive cells were detected in the HFD+En group in comparison with the HFD or ND groups (Figure. 7a). Western blot analyses further indicated that the expression of DCX protein was enhanced in the hippocampus of the HFD+En group when compared with the HFD group (Figure 7b).

## 4. Discussion

In this study, the results demonstrate that maternal HFD increased the activity of nuclear HDAC4 and decreased the expressions of SOX2, PAX6, Ki67 and DCX in the hippocampi of adult female offspring at 12 weeks old. On the other hand, the level of Nestin showed no significant alteration. Suppression of HDAC4 activity by Mc1568, a class II HDAC inhibitor, effectively enhanced these suppressed markers of NSCs lineage and neurogenesis. These results suggest that the suppression of adult neurogenesis in female offspring due to maternal HFD is mediated by HDAC4. The suppression of adult neurogenesis by maternal HFD was effectively reversed by exposure to environmental enrichment in adulthood.

Maternal overnutrition during gestation and lactation has negative effects on hippocampal functions in offspring [41,42,43,44]. In addition, an increasing amount of evidence indicates that females display less efficient spatial competencies than males [45,46]. Whether these effects are gender specific has not yet been fully evaluated. Previous research has indicated that the female hippocampus may be more vulnerable to the negative effects of maternal HFD than that of the male. We therefore focused on the effects of maternal HFD on the hippocampus of female offspring in the present study. Adult neurogenesis serves as one of the critical processes in the maintenance of hippocampal functions in addition to the existing neural circuit [9]. Previous work from our lab demonstrated that maternal HFD suppresses the hippocampal expression of brain-derived neurotrophic factor (BDNF), which is critical for adult neurogenesis in the dentate gyrus [31]. However, the effects of maternal HFD on neurogenesis in adult female offspring remain largely unknown. In this study, we examined the markers of NSC proliferation, NSC differentiation, and neural differentiation in adult female HFD offspring to assess whether they had experienced any impairment in these processes as a result of maternal HFD. In light of the decrease in Ki67, SOX2, PAX6 and DCX, our results suggest that maternal HFD may negatively impact hippocampal adult neurogenesis via effects on NSC division, the transition from NSCs to NPCs, and neuronal differentiation in female offspring from as early as at 12 weeks old. The suppressed adult neurogenesis may provide less support to the existing neuronal circuits, resulting in a lower capacity of learning and memory, which was demonstrated in our previous study [6].

Maternal diet is a critical modulator of offspring epigenetics [47]. HDCA4 is one member of class II HDACs which negatively regulates learning and memory by gene repression [48]. A previous study demonstrated that enhanced HDAC4 mediated the maternal HFD-induced suppression of hippocampal BDNF [6]. On the other hand, HDAC4 degradation promotes the progression of differentiation [49]. There is increasing evidence to suggest that HDAC4 induced by maternal HFD may suppress adult neurogenesis. In our current study, we demonstrate that the increased nuclear activity of HDAC4 in the hippocampi of the HFD group contributed to suppressing NSC proliferation, the transition from NSCs to NPCs and neuronal differentiation. This suppression could be mediated through effects on SOX2, PAX6, and DCX. HDAC4 is involved in removing acetyl groups from N-terminal lysine residues at positions 9, 14, 18, and 23 of histone 3 as well as positions 5, 8, 12, and 16 of histone 4 [50]. It is possible that removal of the acetyl group from these positions by HDAC4 may be the mechanism by which the expression is decreased for Ki67, Nestin, SOX2, PAX and DCX. However, the precise underlying mechanisms of HDAC4-associated suppression of SOX2, PAX6, and DCX needs further evaluation.

Ki67 is a nuclear protein commonly used as an adult neurogenesis marker for cell division and proliferation in the adult hippocampus. Our results indicate that maternal HFD downregulated Ki67 in the hippocampus of adult female offspring. These results implied a reduction in NSC proliferation. Although HDAC inhibition promotes the increase in Ki67 in adult neurogenesis, few studies specifically focus on the regulation of Ki67 expression by HDAC3. In our study, the suppression of HDAC4 nuclear activity through the use of a specific inhibitor effectively enhanced Ki67 expression. This suggests that the increased HDAC4 activity, as a result of maternal HFD, is responsible for mediating Ki67 suppression in the hippocampus of adult female offspring. However, whether this Ki67 suppression is directly or indirectly regulated by HDAC4 needs further investigation.

SOX2 is a transcription factor that is highly expressed in the subgranular zone (SGZ) of NSCs in the adult dentate gyrus [15,16]. SOX2 is frequently used as a marker for NSCs and plays important roles in NSC proliferation and differentiation. In addition, SOX2-positive cells proliferate into a subpopulation of undifferentiated dividing cells, implying their capability for self-renewal in the adult brain [51]. Although growing evidence suggests the involvement of HDACs in neurogenesis, fewer studies have focused on the effect of HDAC4 on SOX2 expression. In this study, we demonstrated that HDAC4 negatively regulated the SOX2 expression in the hippocampus of adult HFD female offspring. In light of these results, the maternal HFD-enhanced HDAC4 activity in the nuclei might contribute to the suppression of adult neurogenesis in the NSC self-renewal stage.

PAX6 interacts with SOX2 in regulating neurogenesis [52]. In adult neurogenesis, the function of PAX6 is critical for the transition of NSCs into NPCs [18]. A simultaneous reduction in PAX6 and SOX2 in the hippocampus implies less progression of NSC differentiation in the hippocampus of HFD female offspring. Currently, few studies have focused on the regulation of PAX6 by HDAC4. Our results demonstrate that increased HDAC4 contributed to the suppression of PAX6 in the HFD group. However, the underlying mechanism of HDAC4 regulation in PAX6 requires further study.

DCX is a protein that promotes microtubule polymerization. During adult neurogenesis, DCX is expressed in postmitotic NPCs and early immature neurons, and is therefore commonly used as a marker for their identification [19]. In the present study, the expression of DCX was suppressed in the hippocampus of the HFD group, and this could effectively be reversed via the inhibition of HDAC4 by Mc568 as well as through exposure to an enriched environment. The suppression of DCX implies fewer newborn neurons in the adult hippocampus of female HFD offspring. The decrease in newborn neurons could be the result of fewer resources for NSC differentiation. The fact that HDAC4 suppresses the number of DCX-positive neurons has been reported in postnatal neurogenesis [53]. Similarly, our results indicate that HDAC4 contributes to the suppression of DCX expression and to the decrement of DCX-positive cells in HFD offspring. Whether the downregulation of DCX by HDAC4 is a cause or a result of fewer newborn neurons needs further investigation.

The provided data imply that maternal HFD-increased HDAC4 activity in the nuclei might suppress NSC self-renewal by SOX2 suppression, decrease NSC differentiation by PAX6 suppression, and impair neuronal differentiation by DCX suppression, resulting in a decrease in adult neurogenesis in adult female offspring. However, whether the HDAC4-associated suppression of neurogenesis is BDNF-dependent or independent still awaits further confirmation.

Environmental enrichment promotes adult neurogenesis [34,54]. In addition, exposure to an enriched environment effectively reduces the level of nuclear HDAC4 in hippocampal neurons, including those in the dentate gyrus [6]. In the present study, the results further indicate that the exposure of HFD offspring to an enriched environment effectively reversed the levels of Ki67-positive cells and DCX-positive cells concurrent with the decrease in HDAC4 activity. These results imply that the exposure to environmental enrichment might increase NSC division and neuronal differentiation through decreased nuclear HDAC4 activity. On the other hand, it has been suggested that the protective effect of environment enrichment is conferred via BDNF [55]. Given that HDAC4 negatively regulates BDNF expression [6,40], it is possible that the mechanism of environmental enrichment that reverses adult neurogenesis could be mediated by the enhancement of hippocampal BDNF. As a result of the limitations of the study design, precise delineation of the underlying mechanisms of the protective effects triggered by environmental enrichment require further investigation.

## 5. Conclusions

In summary, this study demonstrated that maternal HFD during gestation and lactation increased nuclear HDAC4 activity, leading to suppressed adult neurogenesis in adult female offspring through adverse effects on NSC proliferation, the progress of differentiation and neuronal differentiation. Environmental enrichment might be a feasible therapy to reverse the associated adverse effects through the targeting of HDAC4 function.

## Figures and Tables

**Figure 1 ijerph-17-03919-f001:**
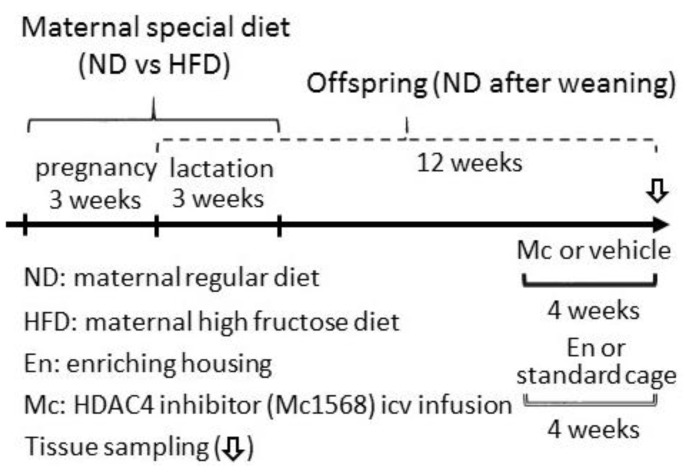
Schematic illustration of the study design. The hippocampal neural stem cell (NSC) proliferation and neuronal differentiation of adult female offspring from a regular maternal diet (ND) or maternal high-fructose diet (HFD) were assayed after 4 weeks of treatment. The treatments of the vehicle or Mc1568 (Mc) intraventricular infusion (icv) were conducted to evaluate the role of maternal diet-altered histone deacetylase 4 (HDAC4) in the adult neurogenesis in female offspring at 12 weeks old. Investigations into the differences between those housed in a standard cage and those housed in enriched housing (En) were conducted to evaluate the therapeutic effect of environmental enrichment on HDAC4-inhibited adult neurogenesis in female offspring at 12 weeks old.

**Figure 2 ijerph-17-03919-f002:**
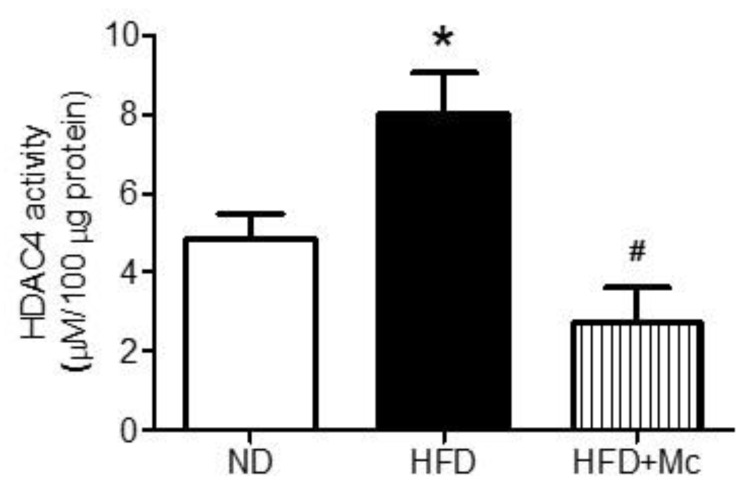
Maternal HFD increased nuclear histone deacetylase 4 (HDAC4) activity in the hippocampus of female offspring at 12 weeks old. Values were mean ± SEM. ND: n = 12; HFD: n = 12; HFD+Mc: n = 6. * *p* < 0.05 indicates significant differences from the ND group and **^#^**
*p* < 0.05 indicates significant differences from the HFD group using a post-hoc Tukey’s multiple range test. ND: female offspring exposed to a maternal normal diet; HFD: female offspring exposed to a maternal high-fructose diet; Mc-1568 (Mc), class II HDAC inhibitor, 5 μM by icv infusion for 4 weeks.

**Figure 3 ijerph-17-03919-f003:**
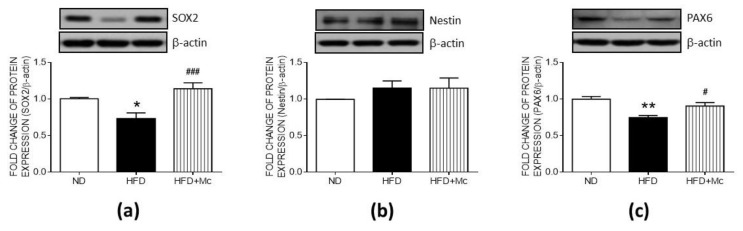
HDAC4 inhibition counteracted the suppression of markers of the hippocampal neural stem cells (NSCs) in the HFD group. The representative gel and densitometric analyses of (**a**) sex-determining region Y box 2 (SOX2); (**b**) Nestin; and (**c**) Paired Box 6 (PAX6) in the hippocampus of female offspring at 12 weeks old. * *p* < 0.05, ** *p* < 0.01 indicate significant differences from the ND group and **^#^**
*p* < 0.05, **^###^**
*p* < 0.001 indicate significant differences from the HFD group using a post-hoc Tukey’s multiple range test. ND: n = 12; HFD: n = 12; HFD+Mc: n = 6. ND: female offspring exposed to a maternal normal diet; HFD: female offspring from maternal high-fructose diet; Mc-1568 (Mc), 5 μM by icv infusion for 4 weeks. β-actin was used as an internal loading control.

**Figure 4 ijerph-17-03919-f004:**
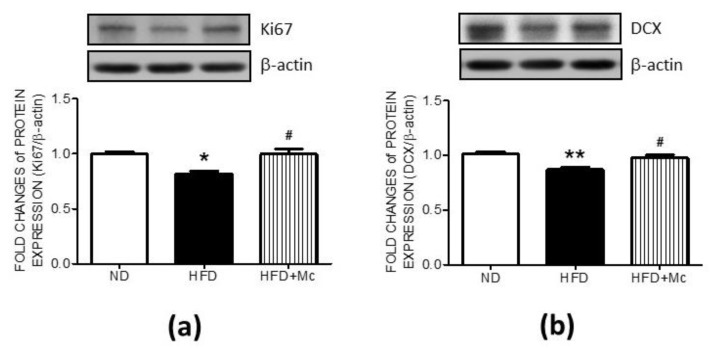
Inhibition of HDAC4 by Mc1568 interfered with the suppression of both Ki67 and DCX expression in the hippocampus of the HFD group. The representative gel and densitometric analyses of (**a**) Ki67, and (**b**) DCX in female offspring hippocampi at 12 weeks old. ** p* < 0.05, *** p* < 0.01 indicate significant differences from the ND group and ***^#^***
*p* < 0.05 indicate significant differences from the HFD group using a post-hoc Tukey’s multiple range test. ND: n = 12; HFD: n = 12; HFD+Mc: n = 6. ND: female offspring exposed to a maternal normal diet; HFD: female offspring exposed to a maternal high-fructose diet; Mc-1568 (Mc), 5 μM by icv infusion for 4 weeks. β-actin was used as an internal loading control.

**Figure 5 ijerph-17-03919-f005:**
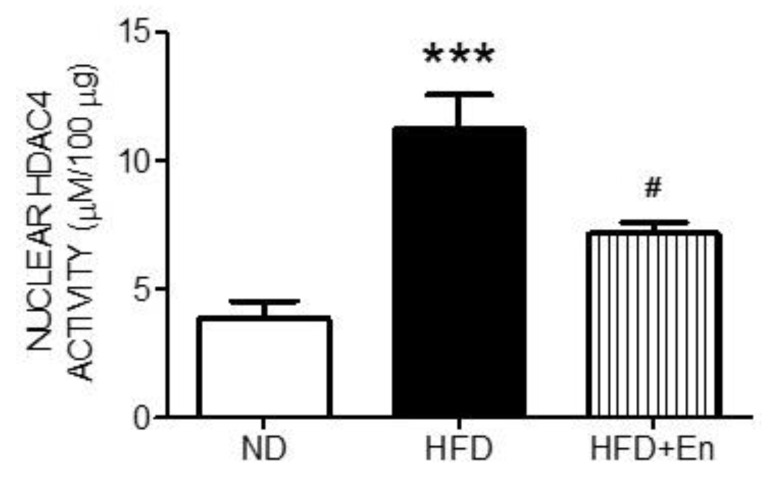
Enriched housing counteracted maternal HFD-increased nuclear HFDAC4 activity in the hippocampus at 12 weeks old. Values were mean ± SEM, ND: n = 12; HFD: n = 12; HFD+En: n = 6. *** *p* < 0.001 indicate significant differences from the ND group and **^#^**
*p* < 0.05 indicate significant differences from the HFD group using a post-hoc Tukey’s multiple range test. ND: female offspring exposed to a maternal normal diet; HFD: female offspring exposed to a maternal high-fructose diet; Mc-1568 (Mc), 5 μM by icv infusion for 4 weeks.

**Figure 6 ijerph-17-03919-f006:**
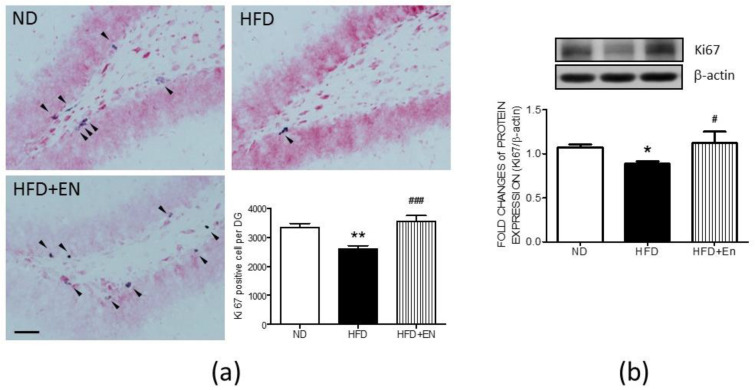
Enriched housing counteracted maternal HFD suppression of Ki67 expression and cell proliferation in the hippocampus of the HFD group. (**a**) Representative images of Ki67-positive cells and cell numbers in the dentate gyrus, and (**b**) protein expression of Ki67 in the hippocampus of female offspring at 12 weeks old. Values were mean ± SEM, ND: n = 12; HFD: n = 12; HFD+En: n = 6. ** p* < 0.05, *** p* < 0.01 indicate significant differences from the ND group and *^#^ p* < 0.05, *^###^ p* < 0.001 indicate significant differences from the HFD group using a post-hoc Tukey’s multiple range test. ND: female offspring exposed to a maternal normal diet; HFD: female offspring exposed to a maternal high-fructose diet; enriched housing for 4 weeks (En). β-actin was used as the internal loading control. Arrowhead: Ki67-positive cells. Scale bar: 50 μm.

**Figure 7 ijerph-17-03919-f007:**
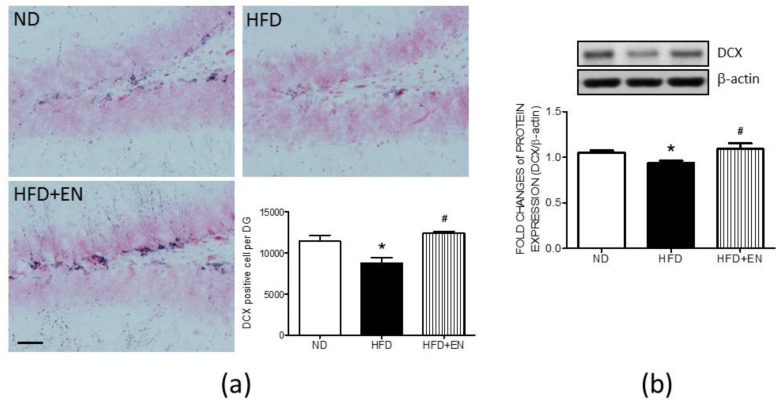
Enriched housing counteracted the suppression of doublecortin (DCX) in the maternal HFD group. (**a**) Representative images of DCX-positive cells and cell numbers in dentate gyrus, and (**b**) protein expression of DCX in the hippocampus of female offspring at 12 weeks old. Values were mean ± SEM, ND: n = 12; HFD: n = 12; HFD+En: n = 6. ** p* < 0.05 indicate significant differences from the ND group and *^#^ p* < 0.05 indicate significant differences from the HFD group using a post-hoc Tukey’s multiple range test. ND: female offspring exposed to a maternal normal diet; HFD: female offspring exposed to a maternal high-fructose diet; enriched housing for 4 weeks (En). β-actin was used as the internal loading control. Scale bar: 50 μm.

**Table 1 ijerph-17-03919-t001:** The dietary contents of regular chow (ND) and high-fructose diet (HFD).

g/kg	Regular Chow (ND)	60% High-Fructose Diet (HFD)
Fructose	-	600
Lard	50	50.0
Casein	232.3	207.0
Cellulose	51	79.81
DL-Methionine	6.7	3.0
Mineral Mix Rogers-Harper (170760)	>7	50.0
Zinc Carbonate		0.04
Vitamin Mix Teklad (40060)		10.0
Food Color (Green)	-	0.15
% kcal from		
Carbohydrate	57.996	66.8
Fat	13.496	13.0
Protein	28.507	20.2
kcal/g	3.35	3.6

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
