# Peer review of "Environmental Stimulation Counteracts the Suppressive Effects of Maternal High-Fructose Diet on Cell Proliferation and Neuronal Differentiation in the Dentate Gyrus of Adult Female Offspring via Histone Deacetylase 4"

_ijerph, 2020, doi:10.3390/ijerph17113919_

Round 1

Reviewer 1 Report

Reviewer comments

The study by Liu et.al, investigated the role of High fructose diet (HFD) on cell proliferation and neuronal differentiation. They also show that the HFD mediated action can be regulated by HDAC inhibitor and environmental cues. This study is interesting but there are many flaws in the experiments. Moreover, the manuscript is written poorly, which makes it difficult to read and understand. The manuscript should undergo major changes before it can be considered for publication.

Major comments

  1. Authors should take help of some language editing team to improve the english. There are many mistakes and the sentences are framed very poorly. For instance, check Line 61. ‘individuals can be permanently shape by the nutrition support during fetal ….’  Line 64. ‘gestation and lactation modulates the brain functions of adult offspring has been documented……’ The manuscript is filled with such poorly structured sentences.
  2. Why only females were considered for the assays? Is there any specific reason not to include males. This should be justified.
  3. Line 145. What do authors mean by ‘in some experiments’, it is better to list them.
  4. Line 158. Histone deacetylase 4 activity assay. Authors should put detail here, how the protein was purified using the Antibody.
  5. Author’s have shown increased activity of HDAC4 (Fig.1). This assay was performed with purified protein, so how come the HFD treated animals are showing increased activity? There is no explanation/discussion on this. Are there some sequence modifications? It will be good if authors analyze the epigenetic changes and also discuss this in detail.
  6. Discussion is poor. Authors need to improve this substantially. As mentioned earlier, the language should also be checked. At the moment it is not easy to understand the paper.
  7. Authors have subjected the animals to inhibitor and enriched housing. However, it is not mentioned if there were any other changes in the animal due to this, especially on body weight.
  8. All the figure legends indicate n=6-12. This is not informative. Authors should put the exact sample size. One option would be to put the sample size on each bar.

Author Response

We thank the Reviewer 1 for his/her insightful critiques and constructive suggestions. We have responded to all comments from the Reviewer and incorporated our responses into the revised manuscript. The newly added contents and modifications are highlighted in blue color and the English edited contents in red color in the revised manuscript. Please see the attachment for our point-by-point responses.

Reviewer 2 Report

I have attached m a more detailed review of your submitted paper. 

Overall, the paper presents good ideas and something of interest in the current climate and western diet. I do have a number of issues that should be addressed to help better present the findings of the paper.

There are some issues with writing and English, which affect the clarity of the paper. 

In the methods section, how cells are counted is not covered anywhere but is included in the results sections. This is an important section of the results to ensure that the measurement of cell number is correct. It may also be good to include the bodyweight of the females at sacrifice to show any group difference, to ensure no effect of maternal diet on adult feeding patterns. 

I would suggest that the authors ensure how results are referenced in the discussion are correct and clear. 

Author Response

We thank the Reviewer 2 for his/her insightful critiques and constructive suggestions. We have responded to all comments from the Reviewer and incorporated our responses into the revised manuscript. The newly added contents and modifications are highlighted in blue color and the English edited contents in red color in the revised manuscript. Please see the attachment of our point-by-point responses.

Round 2

Reviewer 1 Report

This study can be considered for publication now.

This manuscript is a resubmission of an earlier submission. The following is a list of the peer review reports and author responses from that submission.